

# Genome-wide characterization and expression analyses of the *MYB* superfamily genes during developmental stages in Chinese jujube

Ji Qing[1], Wang Dawei[1], Zhou Jun[1,2], Xu Yulan[1], Shen Bingqi[1] and Zhou Fan[1]

[1] Southwest Forestry University, Key Laboratory for Forest Resource Conservation and Utilization in the Southwest Mountains of China, Ministry of Education, Kunming, Yunnan, China
[2] North Minzu University, College of Life Science and Engineering, Yinchuan, China

Corresponding author
Zhou Jun, zhoujunbo@163.com

## ABSTRACT

The MYB transcription factor (TF) superfamily, one of the largest gene superfamilies, regulates a variety of physiological processes in plants. Although many MYB superfamily genes have been identified in plants, the MYB TFs in Chinese jujube (*Ziziphus jujuba* Mill.) have not been fully identified and characterized. Additionally, the functions of these genes remain unclear. In total, we identified 171 *MYB* superfamily genes in jujube and divided them into five subfamilies containing 99 genes of the R2R3-MYB subfamily, 58 genes of the MYB-related subfamily, four genes of the R1R2R3-MYB subfamily, one gene of the 4R-MYB subfamily, and nine genes of the atypical MYB subfamily. The 99 R2R3-MYB genes of jujube were divided into 35 groups, C1–C35, and the 58 MYB-related genes were divided into the following groups: the R-R-type, CCA1-like, I-box-binding-like, TBP-like, CPC-like, and Chinese jujube-specific groups. *ZjMYB* genes in jujube were well supported by additional highly conserved motifs and exon/intron structures. Most R1 repeats of MYB-related proteins comprised the R2 repeat and had highly conserved EED and EEE residue groups in jujube. Three tandem duplicated gene pairs were found on 12 chromosomes in jujube. According to an expression analysis of 126 *ZjMYB* genes, MYB-related genes played important roles in jujube development and fruit-related biological processes. The total flavonoid content of jujube fruit decreased as ripening progressed. A total of 93 expressed genes were identified in the RNA-sequencing data from jujube fruit, and 56 *ZjMYB* members presented significant correlations with total flavonoid contents by correlation analysis. Five pairs of paralogous *MYB* genes within jujube were composed of nine jujube *MYB* genes. A total of 14 *ZjMYB* genes had the same homology to the *MYB* genes of *Arabidopsis* and peach, indicating that these 14 *MYB* genes and their orthologs probably existed before the ancestral divergence of the MYB superfamily. We used a synteny analysis of *MYB* genes between jujube and *Arabidopsis* to predict that the functions of the *ZjMYBs* involve flavonoid/phenylpropanoid metabolism, the light signaling pathway, auxin signal transduction, and responses to various abiotic stresses (cold, drought, and salt stresses). Additionally, we speculate that *ZjMYB108* is an important TF involved in the flavonoid metabolic pathway. This comprehensive analysis of *MYB* superfamily genes in jujube lay a solid foundation for future comprehensive analyses of *ZjMYB* gene functions.

## INTRODUCTION

Transcription factors (TFs) play essential roles in plants by controlling the expression of genes, activating or inhibiting the transcription of other genes, or interacting with other TFs to regulate gene transcription (*Singh, Foley & Onate-Sanchez, 2002*; *Liu, White & MacRae, 1999*). The MYB family is a large TF family present in all eukaryotes, and MYBs regulate a variety of physiological processes in plants (*Riechmann et al., 2000*). The first identified MYB gene was *v-myb* in 1982 from the avian myeloblastosis virus, a chicken oncogene that leads to acute myeloblastic leukemia (*Klempnauer, Gonda & Bishop, 1982*), thus leading to the name *myb*. C-*myb,* a v-MYB-related gene was subsequently found in animal cells. Corresponding genes were also identified in human tumor cells (A-MYB and B-MYB) at the same time and these genes were found to modulate cell proliferation, differentiation, and apoptosis (*Weston, 1998*). The first identified *MYB* gene of a plant was C1, which was isolated from *Zea mays* (*Paz-Ares et al., 1987*) and is involved in anthocyanin biosynthesis. As the *Arabidopsis MYB* gene family has gradually been identified (*Stracke, Werber & Weisshaar, 2001*; *Yanhui et al., 2006*), more *MYB* genes have been identified in many other plants (*Zhang et al., 2018b*; *Li et al., 2016b*; *Hou et al., 2014*; *Zhou et al., 2015*); these genes are identified by the family-specific feature of a highly conserved MYB domain at the N-terminus (*Lipsick, 1996*; *Mmadi et al., 2017*; *Dubos et al., 2010*). This domain gene usually comprises one to four imperfect amino acid sequence repeats (R1–R4) of approximately 50–53 amino acids (*Dubos et al., 2010*; *Mmadi et al., 2017*), with each forming three α-helices, and the second and third helices form a helix-turn-helix (HTH) motif (*Ogata et al., 1996*; *Dubos et al., 2010*). *MYB* genes are classified into the following subfamilies according to the number of MYB imperfect tandem repeats (Rs) of the proteins: MYB-related (or 1R-MYB, one R), R2R3-MYB (two Rs), R1R2R3-MYB (three Rs), and 4R-MYB (four Rs) (*Dubos et al., 2010*; *Zhang et al., 2018b*).

Many studies have shown that MYB TFs are involved in physiological and biochemical processes in plants and responses to various biotic as well as abiotic stresses (*Abe et al., 2003*; *Agarwal et al., 2006*; *Cominelli & Tonelli, 2009*; *Raffaele et al., 2008*), and the function of MYB has also been studied in detail in some MYB proteins. For example, *MYB7* plays a role in kiwifruit (*Actinidia chinensis*) through transcriptional activation of metabolic pathway genes to modulate carotenoid and Chl pigment accumulation in tissues (*Ampomah-Dwamena et al., 2018*). The *AtDIV2* of the R-R-type *MYB* gene in *Arabidopsis* is required for ABA signaling and plays a negative role in salt stress (*Fang et al., 2018*). In tobacco, overexpression of the *PbrMYB5* gene enhanced tolerance to chilling stresses (*Xing et al., 2018*). In tomato (*Solanum lycopersicum* L.) plants, the R3-MYB gene is involved in a feedback mechanism, and if it is activated by endogenous or exogenous stimuli, anthocyanin production is inhibited (*Colanero, Perata & Gonzali, 2018*). During continuous light treatments, the levels of *McMYB10* increased and promoted the

expression levels of *McCOP1-1* and *McCOP1-2* as well as anthocyanin biosynthesis in crabapple (*Li et al., 2018*). MYB TFs are also involved in flavonoid/phenylpropanoid metabolism; for example, overexpression of the *AtMYB12* gene of *Arabidopsis* enhanced the accumulation of flavonoid content under low temperatures in a light-dependent manner (*Bhatia et al., 2018*). During strawberry ripening, the *MYB10* gene in *Fragaria* × *ananassa* plays a general regulatory role in the flavonoid/phenylpropanoid pathway (*Puche et al., 2014*). In addition, *ATMYB12* (*Bhatia et al., 2018*), *ATMYB018* (*Ballesteros et al., 2001*), *ATMYB21* (*Shin et al., 2002*), *ATMYB075*, and *ATMYB090* (*Li et al., 2006*) genes are involved in the light signaling pathway in *Arabidopsis*. R2R3-MYB TF family members in petunia (*Petunia hybrida*) are developmentally and environmentally regulated to control complex floral and vegetative pigmentation patterning (*Albert et al., 2011*).

Chinese jujube (*Ziziphus jujuba* Mill.) is a traditional economic tree species in China. Jujube has flourished for a long time because of its strong resistance, simple management, high yield, rich nutrition, and good economic and ecological benefits (*Li et al., 2007*; *Zhao, Liu & Tu, 2008*). Many consumers favor jujube because of its good taste, rich nutrition, powerful health functions, and alternative medicinal properties (*Wang et al., 2018b*; *Zhang et al., 2018a*; *Lam et al., 2016*). Fresh jujube fruits develop rapidly particularly in the high-quality cultivar "Dongzao" (*Yuan et al., 2017*). The entire Chinese jujube genome sequence was obtained in 2014 (*Liu et al., 2014*), providing data for genome-wide analyses of the MYB superfamily. Although many studies have shown that MYB TFs are involved in physiological and biochemical processes in plants, the MYB TFs of jujube have not been fully identified and characterized, and the expression of *MYB* genes in the different developmental stages of fresh jujube fruit remains unclear. Thus, in this study, the MYB superfamily members in jujube were analyzed and identified, and their protein physicochemical properties, motif and exon/intron composition, correlation with flavonoid content, syntenic relationships, and expression levels in different developmental stages of fresh jujube were analyzed. These findings should inform the characterization of *ZjMYB* genes, and this study will be helpful for future functional studies of the *ZjMYB* superfamily genes involved in fruit development in jujube.

## MATERIALS AND METHODS

### Plant materials

A total of 12-year-old Chinese jujube served as test materials at the "Xuefeng" ecological park of Gengjiaying township, Yiliang county in Kunming city of Yunnan province, China, which is the teaching experiment base of Southwest Forestry University. Fruits were selected from five periods, including the young stage, enlargement stage, white mature stage, half-red stage, and full-red stage. The samples of these five periods were harvested in turn 25, 39, 74, 83, and 99 days after anthesis. These fruits were immediately frozen in liquid nitrogen and stored at −80 °C. Three biological repeats were performed for each developmental stage for RNA-sequencing, total flavonoid content determination, and qRT-PCR analysis.
## Jujube *MYB* superfamily gene identification

The completed jujube genome sequence and chromosome information were identified from the DDBJ/EMBL/Gen-Bank (accession JREP00000000), which provides Chinese jujube genome data (*Liu et al., 2014*). The Pfam database (http://pfam.xfam.org/) was searched to obtain the hidden Markov model (HMM) profile for the MYB binding domain (PF00249) (*Finn et al., 2016*; *Zhang et al., 2018b*), and all putative MYB genes were obtained from the jujube genome database (*Liu et al., 2014*). The presence of a MYB domain in the selected MYB proteins was further verified using the online program SMART (http://smart.embl-heidelberg.de/) and HMMER (https://www.ebi.ac.uk/Tools/hmmer/). Next, manual analysis was performed using the ClustalX program to confirm MYB conserved domains or motifs. Ultimately, we confirmed that the genes containing the MYB domain were members of the MYB superfamily. Based on previous research, we downloaded 198 MYB family protein sequences and chromosome information in *Arabidopsis* from the *Arabidopsis* information resource (TAIR) (http://www.Arabidopsis.org/) (*Stracke, Werber & Weisshaar, 2001*; *Yanhui et al., 2006*).

## Multiple sequence alignment and phylogenetic tree construction of MYB proteins

In this study, we will refer to the classification of the *Arabidopsis* MYB gene family and the results of a phylogenetic tree for subfamily classification of the jujube MYB family and grouping (*Dubos et al., 2010*; *Yanhui et al., 2006*; *Zhang et al., 2018b*). We constructed the first phylogenetic tree with 126 R2R3-MYB proteins, five R1R2R3-MYB proteins, and one 4R-MYB protein of *Arabidopsis* and 171 MYB family proteins of jujube. We used the ClustalW program for multiple comparisons of amino acid sequences. The neighbor-joining phylogenetic tree was constructed by 1,000 repeated bootstrap analyses using MEGA 6 software (*Tamura et al., 2013*). Genes not belonging to R2R3-MYB, R1R2R3-MYB and 4R-MYB in the first phylogenetic tree of MYB genes were used to construct a second phylogenetic tree with the 60 1R-MYB proteins and six atypical MYB proteins in *Arabidopsis*. Phylogenetic tree construction parameters reference the first phylogenetic tree.

## Gene structure and conserved motif analysis

The exon-intron structures of *ZjMYB* genes were generated with the gene structure display server (GSDS: http://gsds.cbi.pku.edu.cn/) (*Hu et al., 2015*). The conserved motifs of *ZjMYB* proteins were defined using the MEME program (http://meme-suite.org/tools/meme) (*Bailey et al., 2009*). The following parameter settings were used: distribution of motifs, zero or one per sequence; the minimum width and maximum width of motifs, six and 250, respectively; maximum number of motifs to find, 15; and default parameters. The conserved domains of *ZjMYB* proteins were defined using Pfam (http://pfam.xfam.org/). We visualized the Pfam, MEME, and GSDS results using TBTOOLS software (https://github.com/CJ-Chen/TBtools) (*Chen et al., 2018*; *Liu et al., 2017a*). The ExPASy online tool (http://www.expasy.ch/tools/protparam.html) was used to analyze the physiological and biochemical characteristics of the *ZjMYB* genes.

## Distribution on the chromosome and tandem duplication analysis of *MYB* genes in jujube

The genome location of each *ZjMYB* member and the length of each chromosome of jujube was obtained from the Chinese jujube database (*Liu et al., 2014*). Based on the above information, the *ZjMYB* genes were mapped to corresponding locations on the 12 chromosomes using the circle gene viewer package of TBtools (*Krzywinski et al., 2009*; *Chen et al., 2018*). Tandem duplication analysis of *ZjMYB* genes was performed using the MCScanX tool (*Wang et al., 2012*).

## Analysis of *ZjMYB* gene expression from RNA-Seq data

RNA-Seq reads were obtained with an Illumina HiSeq 2000. The fragments per kilobase of exon per million mapped reads (FPKM) values were calculated based on RNA-Seq reads. The heatmap was generated with TBTOOLS software (*Chen et al., 2018*; *Liu et al., 2017a*); the color scale shown represents FPKM counts, and the ratios were log2 transformed. To confirm the transcriptome data, we chose 10 *MYB* genes and quantified them using quantitative real-time PCR experiments. Total RNA was extracted using the Plant RNA Kit (TaKaRa Biotechnology Co. Ltd., Dalian, China). DNA-free RNA was used for synthesis of the first strand of cDNA by using the Prime Script II 1st Strand cDNA Synthesis Kit (TaKaRa Biotechnology Co. Ltd., Dalian, China) per the manufacturer's recommendations. Quantitative RT-PCR was carried out with the Rotor-Gene Qreal-time PCR system (Qiagen, Hilden, Germany) instrument using Fast Super EvaGreen qPCR Master Mix (US Everbright Inc., Suzhou, China). The gene-specific primers of the 10 ZjMYB genes were designed using Primer Premier 5.0 software (*Lalitha, 2000*), and the specific primer pairs are listed in Table S1. The UBQ gene of jujube was used as an internal control. Each reaction contained 10 μL of Fast Super EvaGreen qPCR Master Mix (US Everbright Inc., Suzhou, China), 0.4 μL of each primer, two μL of cDNA, and 7.2 μL of $H_2O$ for a final volume of 20 μL according to the manufacturer's instructions. The reaction was carried out as follows: 95 °C for 2 min, followed by 40 cycles of 95 °C for 5 s, 60 °C for 5 s, 72 °C for 25 s. Each reaction was performed using three-step amplification with three technical replicates, and the data from qRT-PCR amplification were analyzed using the $2^{-\triangle\triangle CT}$ method (*Livak & Schmittgen, 2001*).

## Total flavonoid determination

The determined total flavonoid content was based on the colorimetric assay method at 510 nm, with slight modifications (*Kou et al., 2015*; *Yu et al., 2012*). Briefly, different concentrations (8, 16, 20, 32, 48, 64, and 80 mg/L) of a standard solution of rutin or extract was mixed with 0.3 mL of 5% $NaNO_2$. The solution was left to stand for 6 min, and then 0.3 mL of 10% $Al(NO_3)_3$ was added. After 6 min, four mL of 4% NaOH was added. Then, the mixture was made to 10 mL with distilled water. The absorbance of the final mixture was measured at 510 nm against a prepared blank using a spectrophotometer. The total flavonoid content was analyzed using OriginPro 8.5 software.

## Correlation analysis

We performed a correlation analysis of the original FPKM counts to determine the expression levels of MYB superfamily genes with the total flavonoid content. Pearson correlation analysis was performed using the corr.test function of R software, and the coroplot software package of R was used for significance tests and graphing. $P < 0.05$ was considered significant.

### *ZjMYB* gene synteny analysis

Synteny analysis of *ZjMYB* genes between jujube and two plant species (peach and *Arabidopsis*) was performed using the MCScanX tool (*Wang et al., 2012*). A syntenic analysis map was constructed using TBtools software (*Chen et al., 2018*; *Liu et al., 2017a*). The protein sequences and the genome location of each MYB gene and the length of each chromosome of *Arabidopsis* were downloaded from the TAIR database (http://www.arabidopsis.org/) (*Swarbreck et al., 2008*). Peach protein sequences and the genome location of each MYB gene and the length of each chromosome were downloaded from the phytozome database (https://phytozome.jgi.doe.gov/) (*Zhang et al., 2018b*).

## RESULTS

### Identification of jujube *MYB* genes and analysis of their protein physicochemical properties

A total of 171 *ZjMYB* genes with typical MYB or MYB-like domains were selected from the jujube genome database on the basis of the HMM profile of the MYB domain. The corresponding chromosomal locations were provisionally ordered by name such that the *MYB* genes were named *ZjMYB1* through *ZjMYB171*. The geneIDs, gene lengths, and physiological and biochemical characteristics of the *ZjMYB* genes are listed in Table S2. The lengths of the coding sequences (CDSs) of the *ZjMYB* genes ranged from 276 to 3,645 bp. The lengths of the protein sequences of *ZjMYB* genes ranged from 91 to 1,214 amino acids. The isoelectric point values for *ZjMYB* proteins ranged from 4.78 (*ZjMYB108*) to 10.39 (*ZjMYB145*), with an average of 7.21, which is similar to those for MYB proteins in sesame (*Sesamum indicum* L.) and peach (*Mmadi et al., 2017*; *Zhang et al., 2018b*). The molecular weight ranged from 10,558.01 Da (ZjMYB74) to 136,172.41 Da (ZjMYB23), with an average of 42,499.15 Da. The GRAVY (grand average of hydropathy) average value was −0.74, which is similar to that for peach MYB proteins (*Zhang et al., 2018b*). The most likely homologous gene of each jujube MYB gene in *Arabidopsis* was obtained from the BLAST tool in the TAIR database (Table S2).

### Phylogenetic trees and group classification of *ZjMYB* proteins

This study identified 171 *MYB* genes in jujube. As shown in Figs. 1 and 2, we found 58 MYB-related proteins, 99 R2R3-MYB proteins, four R1R2R3-MYB proteins, one 4R-MYB protein, and nine atypical MYB proteins. R2R3-MYB proteins accounted for 58% of the *ZjMYB* proteins as the largest subfamily, and 4R-MYB subfamily proteins accounted for 0.6% as the smallest subfamily. MYB-related proteins accounted for 40% of *ZjMYB* proteins as the second largest subfamily.

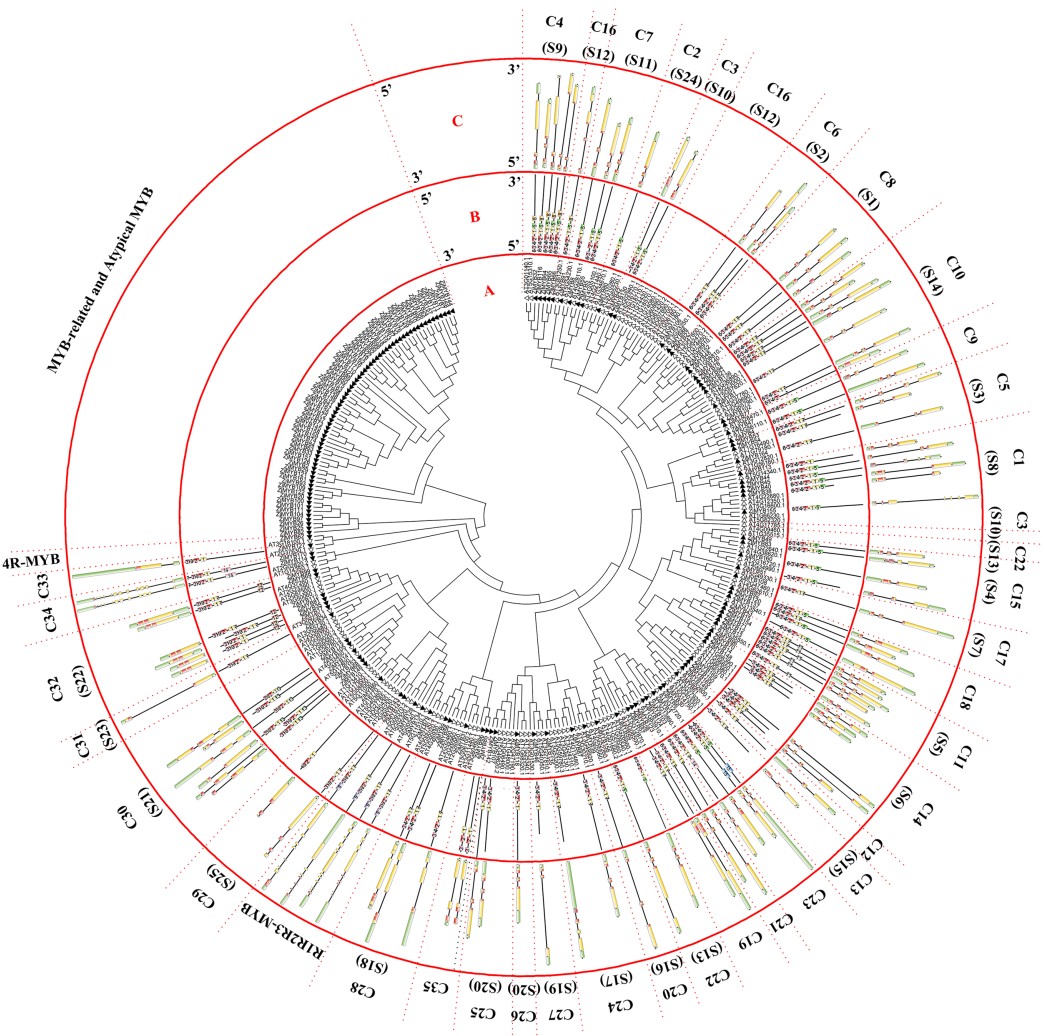

**Figure 1 Phylogenetic tree of the R2R3-MYB, R1R2R3-MYB, and 4R-MYB subfamilies of jujube and *Arabidopsis* and conserved motif and gene structure analysis of the R2R3-MYB and R1R2R3-MYB subfamily proteins of jujube.** (A) Phylogenetic tree of the MYB proteins of jujube and *Arabidopsis*. The sequences of the 171 MYB superfamily proteins of jujube and 132 *Arabidopsis* R2R3-MYB, R1R2R3-MYB, and 4R-MYB proteins were aligned by ClustalW, and the phylogenetic tree was constructed using MEGA 6. The small white triangles represent the 171 jujube MYB proteins, and the small black triangles represent the 132 *Arabidopsis* MYB proteins. The names of each group are marked by English letters with Arabic numbers. (B) Distributions of conserved motifs in ZjMYB genes. The motifs of numbers 1–15 are indicated in different colored boxes. The sequence information of the motifs is provided in Table S2. (C) The exon–intron structure of jujube MYB genes. The yellow boxes and black lines indicate exons and introns, respectively. The green boxes at the two ends of the sequences indicate upstream and downstream regions, respectively. The MYB domains are highlighted by red boxes.

The first phylogenetic tree (Fig. 1A) comprised 171 *ZjMYB* superfamily genes and 126 R2R3-MYB genes, five R1R2R3-MY*B* genes, and one 4R-MYB gene of *Arabidopsis*. A total of 58 MYB-related proteins and nine atypical MYB proteins of the *ZjMYB* superfamily were clustered alone. Then, we used those genes with 60 MYB-related and six atypical MYB proteins of *Arabidopsis* to construct the second phylogenetic tree (Fig. 2A).

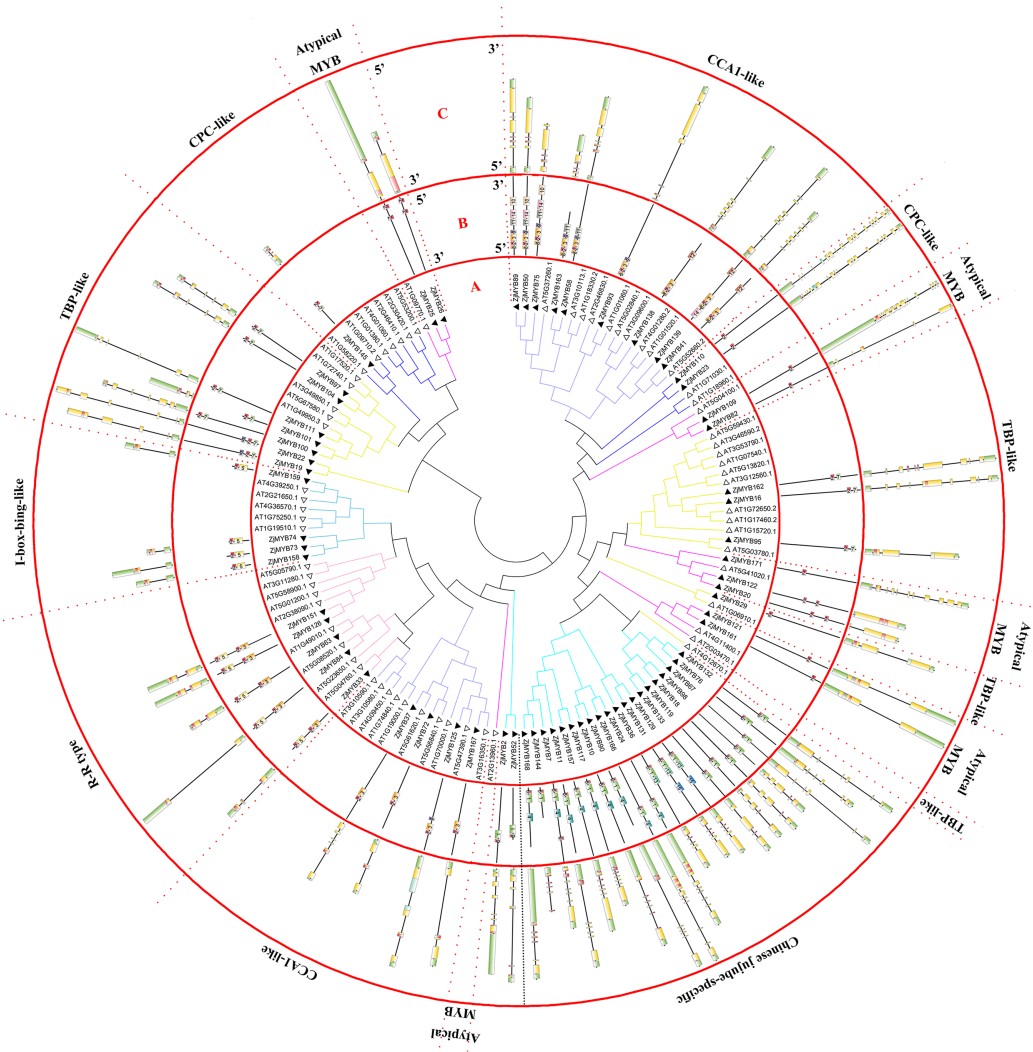

**Figure 2 Phylogenetic tree of the MYB-related (1R-MYB) and atypical MYB subfamilies of jujube and *Arabidopsis*, and conserved motif and gene structure analysis of MYB-related and atypical MYB subfamily proteins of jujube.** (A) Phylogenetic tree of the MYB proteins of jujube and *Arabidopsis*. The sequences of the 58 MYB-related and nine atypical MYB proteins of jujube with the 60 MYB-related and six atypical MYB proteins of *Arabidopsis* were aligned by ClustalW, and the phylogenetic tree was constructed using MEGA 4. The small white triangles represent the 67 jujube MYB proteins and the small black triangles represent the 66 *Arabidopsis* MYB proteins. The names of each group are marked by English letters with Arabic numbers. (B) Distributions of conserved motifs in ZjMYB genes. The motifs of numbers 1–15 are indicated in different colored boxes. The sequence information of the motifs is provided in Table S3. (C) The exon–intron structure of jujube MYB genes. The yellow boxes and black lines indicate exons and introns, respectively. The green boxes at the two ends of the sequences indicate upstream and downstream regions, respectively. The MYB domains are highlighted by red boxes.

As shown in Fig. 1A, one jujube 4R-MYB protein (ZjMYB62) was clustered into the 4R-MYB protein of *Arabidopsis*, four jujube R1R2R3-MYB proteins were clustered into five R1R2R3-MYB proteins of *Arabidopsis*, and 99 jujube R2R3-MYB proteins were clustered into 126 R2R3-MYB proteins of *Arabidopsis*. These 99 R2R3-MYB genes of jujube were divided into 35 groups, C1–C35, based on the topology of the tree and the

classification of the MYB superfamily in *Arabidopsis*, peach and pear (*Pyrus bretschneideri*) (*Dubos et al., 2010*; *Zhang et al., 2018b*; *Li et al., 2016b*). The number of members in each group ranged from one to seven in jujube. C10 had seven members and was the largest group.

In the second phylogenetic tree (Fig. 2A), 58 MYB-related (1R-MYB) proteins were divided into six groups based on the topology of the tree and their classifications in *Arabidopsis* and peach (*Yanhui et al., 2006*; *Zhang et al., 2018b*). CPC-like (three members), TBP-like (11 members), I-box-binding-like (four members), R-R-type (five members), CCA1-like (13 members), and Chinese jujube-specific (22 members) groups were present.

TBP-like was the second largest group among the six groups of the MYB-related subfamily, as in peach (*Zhang et al., 2018b*). The 22 members did not cluster with any *Arabidopsis* proteins. However, this result has been shown in previous studies in peach (*Zhang et al., 2018b*) and sweet orange (*Hou et al., 2014*), and the authors included those genes in the MYB-related (1R-MYB) subfamily. The author of the peach study reported that those genes might have fruit-related functions and were either not needed in *Arabidopsis* or were obtained after divergence from the last common ancestor (*Zhang et al., 2018b*). This study further contributes to this speculation as the function of those genes remains unknown. Nine atypical MYB proteins were clustered in four branches in the phylogenetic tree.

## *ZjMYB* gene structure and protein motif analysis

The MEME program predicted 15 conserved motifs of the MYB proteins of jujube, which appeared in R2R3-MYB and R1R2R3-MYB (Table S3; Fig. 1B), MYB-related, and atypical MYB (Table S4; Fig. 2B), and 4R-MYB (Table S5). The gene exon/intron structures were analyzed in R2R3-MYB with R1R2R3-MYB (Table S3; Fig. 1C), MYB-related with atypical MYB (Table S4; Fig. 2C), and 4R-MYB.

R2R3-MYB and R1R2R3-MYB had three identical highly conserved motifs (Fig. 1B), motif 3, motif 2, and motif 1. A total of 78 of the 99 R2R3-MYB proteins had four identical highly conserved motifs, motif 3, motif 4, motif 2, and motif 1. As the sequence logos of the R2R3-MYB protein repeats show (Fig. 3), motif 3, motif 4, and the front part of motif 2 constitute the R2 repeat in jujube and include a highly conserved separated triplet of Tryptophan (W) to maintain the HTH structure. This result was consistent with those of previous studies (*Zhang et al., 2018b*; *Hou et al., 2014*; *Yanhui et al., 2006*) showing that the back parts of motif 2 and motif 1 constituted the R3 repeat. In this study, the first Tryptophan (W) of the R3 repeat in jujube was replaced by *Leu*cinol hydrochloride (L), Isoleucine (I), *Phe*nylalanine (F), Tyrosine (Y), and Methionine (M) (the back part of motif 2 in Fig. 3). The second and third Tryptophan residues were conserved in most R3 repeats in this study; only one (ZjMYB170) occurred in group C34 (two members), and the third Tryptophan (W) residue was replaced by *Phe*nylalanine (F) (the last amino acid of motif 1 in Fig. 3). Consistent with previous research, the R2 repeat also had the following highly conserved groups of EED residues (motif 3): glutamic acid (E)-glutamic acid (E)-aspartic acid (D) and EEE residues (motif 2) (*Zhang et al., 2018b*; *Li et al., 2016b*).

R2 repeat

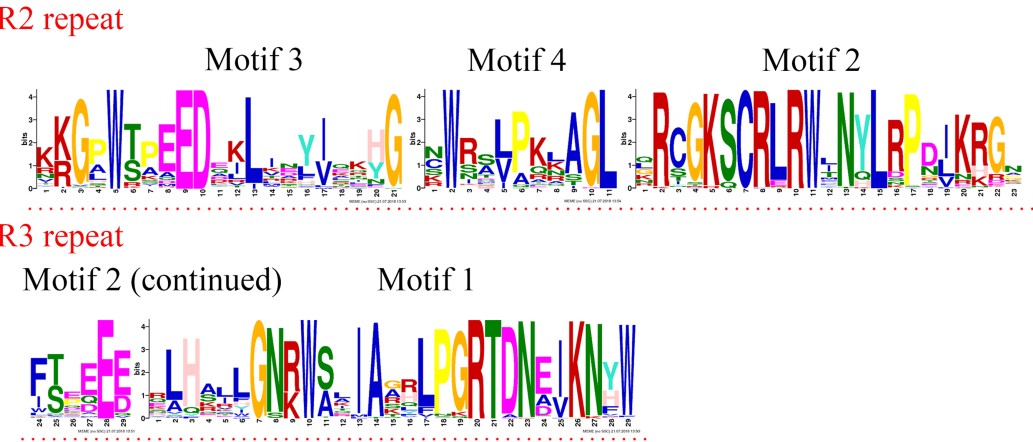

Motif 3          Motif 4          Motif 2

R3 repeat

Motif 2 (continued)          Motif 1

**Figure 3 R2 and R3 MYB repeats of the proteins of R2R3-MYB subfamily in jujube.** The logo sequences of R2 and R3 repeats were composed of motifs 3, 4, 2, and 1 in jujube. The overall height of each stack showed the conservation of the MYB protein sequence at that position. English letters indicate the different type of amino acid residue.               

In jujube, the remaining residues were highly conserved in the R2 repeat; the eighth and 16th residues behind the first Tryptophan (W) were leucine (L) and glycine (G), and the eighth (G) and ninth (L) residues were behind the second Tryptophan (W). In front of the third Tryptophan (W) in the R2 repeats, the following nine consecutive highly conserved residues were observed: *arg*inine (R), *cys*teine (C), Glycine (G), Lysine (K), Serine (S), *cys*teine (C), *arg*inine (R), leucine (L), and *arg*inine (R). This characteristic was also found in other plant species, such as peach (*Zhang et al., 2018b*) and Chinese white pear (*Li et al., 2016b*). Although the authors did not mention this characteristic in their articles, we found it by reanalyzing the data they presented. Other R2R3-MYB proteins (21 members) did not contain motif 4, and they were divided into eight groups, including one in group C7 (three members) and all members in group C29–C35.
A total of 57 of the 99 R2R3-MYB proteins included motif 3, motif 4, motif 2, and motif 1 and also included motif 6. Although ZjMYB113 did not have motif 5 and ZjMYB42 did not have motif 6, groups C1, C2, C3, C9, and C7 had motifs 6, 3, 4, 2, 1, and 5. Groups C16 and C4 had motifs 6, 3, 4, 2, 1, 5, and 8. Motif 8 had highly conserved Tryptophan (W) residues, and the function of this W requires further research. The members of groups C11, C32, C30, and C34 had unique motifs, including 11, 12, 13, and 14 in jujube R2R3-MYB proteins.

Four R1R2R3-MYB proteins contained motifs 9, 3, 10, 2, 1, and 7. As the sequence logos show (Fig. 4), motif 9 constituted the R1 repeat; motifs 3, 10, and 2 constituted the R2 repeat; and motifs 1 and 7 constituted the R3 repeat. The R2 repeat differed between R2R2-MYB and R1R2R3-MYB. Motifs 3, 4, and 2 constituted the R2 repeat of R2R3-MYB.

The CDSs of all R2R3-MYBs were disrupted by introns (Fig. 1C), except for group 32. Most CDSs had three exons and two introns, and all of these exons included two short exons and one relatively long exon, similar to those in peach (*Zhang et al., 2018b*). All group 32 members had no introns and only one exon. However, these members had green regions upstream and downstream of the front and back end of the sequences,

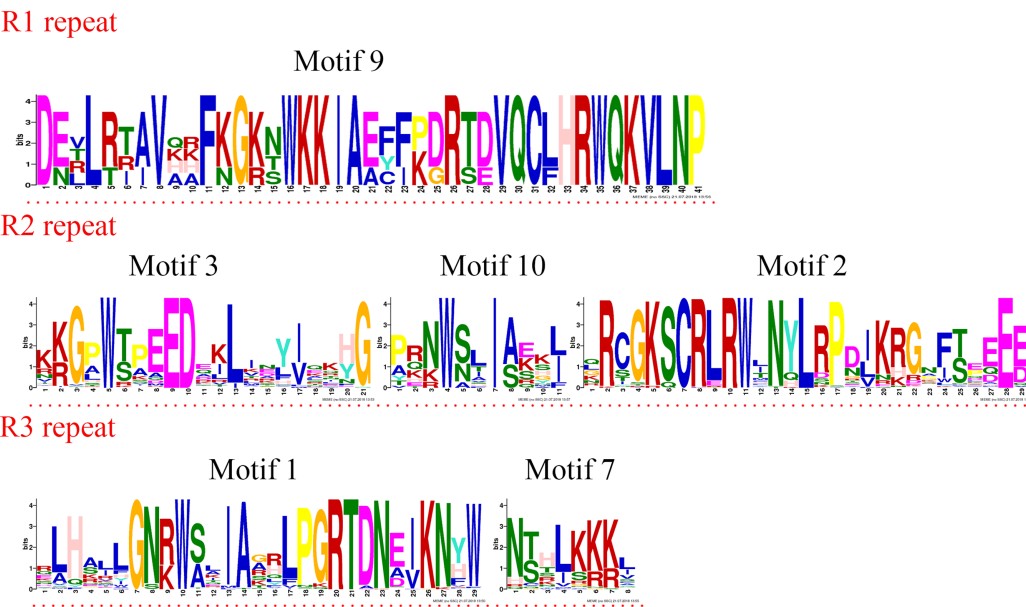

**Figure 4  R1, R2, and R3 MYB repeats of the proteins of R1R2R3-MYB subfamily in jujube.** The logo sequences of R1, R2, and R3 repeats were composed of motifs 9, 3, 10, 2, 1, and 7 in jujube. The overall height of each stack showed the conservation of the MYB protein sequence at that position. English letters indicate the different type of amino acid residue.   

which was consistent with those in peach. Two members of group C34 had a maximum of 11 introns in the R2R3-MYB subfamily. The members of groups 18 and 19 had two exons and one intron. The gene length, exon number and exon/intron structure within the same group of R2R3-MYB were similar, which supported the classification of R2R3-MYB subfamilies. R1R2R3-MYB introns ranged from 6 to 12, and ZjMYB134 had a maximal number of 12 introns.

Although comprehensive analyses of the motifs of R2R3-MYB proteins haves been performed in previous studies, a similar comprehensive motif analysis for the MYB-related subfamily is lacking. In this study, 15 motifs of 58 MYB-related proteins and nine atypical MYB proteins are shown in Fig. 2B. The groups contained a highly conserved motif 2 except the Chinese jujube-specific group. The specific group consisted of at least conserved motifs 1 and 2, which together comprised the R1 repeat. The members of the R-R-type group had motifs 2, 5, 2, and 3 in that order and contained two repeats. Motifs 2 and 5 comprised the first repeat, and motifs 2 and 3 comprised the second repeat. The CCA1-like group was divided into two parts consisting of at least conserved motifs 2 and 3, which together comprised the R1 repeat. All CCA1-like members contained the conserved motif SHAQK with motif 3 of the MYB repeat, consistent with previous studies in *Arabidopsis* (*Yanhui et al., 2006*). Nine members of the second part of the CCA1-like group (12 members) also contained motif 8. Four members of the I-box-like group had motifs 2 and 5, which together constituted the R1 repeat of the I-box-like group. The lengths of all I-box-binding-like protein sequences were short and similar. The 11 members of TBP-like had motifs 2 and 7 except for *ZjMYB29*. Five members of

## R-R (A) was the first repeat

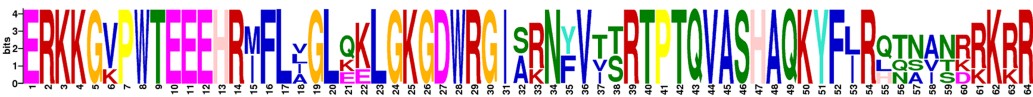

## R-R (B) was the second repeat

## CCA1-like (clade II)

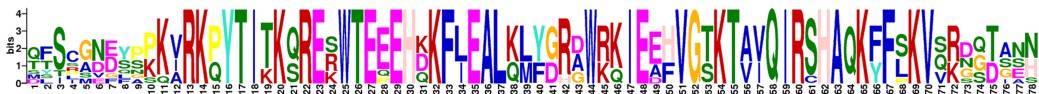

## CCA1-like (clade I)

## TBP-like

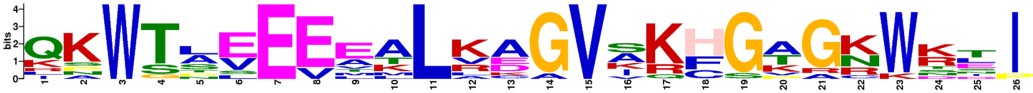

**Figure 5 This logo indicates the sequence similarities of the ZjMYB repeats of the R-R-type, CCA 1-like, and TBP-like MYB-related proteins.**

nine atypical MYB proteins had only motif 2, and other atypical MYB proteins had motifs 2 and 2 in that order, and they comprised an imperfect R1 repeat.

To clearly display the repeats of the seven different groups in MYB-related proteins, we used the MEME program to define the motifs of seven groups with highly conserved motifs (Fig. 5; Table S6). The motifs in the sequence logos of R-R-type repeats had higher similarity than did those in *Arabidopsis* and peach (*Yanhui et al., 2006*; *Zhang et al., 2018b*). The CCA1-like subfamily is composed of two main clades and is the same in *Arabidopsis* (*Yanhui et al., 2006*). The primary structure of clade I repeat DNA-binding domains was (-W-(X 19)-W-), and the repeat of clade II was (-W-(X 18)-W-). The primary structure of the TBP-like repeat DNA-binding domains was (-W-(X 19)-W-). Most R1 repeats of MYB-related proteins contained highly conserved groups of EED or EEE residues in jujube.

The CDSs of the MYB-related subfamily were also disrupted by introns (Fig. 2C). The R-R-type and I-box-binding-like genes had two exons and one intron except for ZjMYB33 of the R-R type, which was similar to groups 18 and 19 of the R2R3-MYB subfamily. Most Chinese jujube-specific genes had four and five introns except for three genes that containing six introns and one that had three introns. On average, the CCA1-like group was disrupted by 4.8 introns, the CPC-like group had 13 introns, the atypical MYB group had 4.1 introns, and the TBP-like group had 4.8 introns.

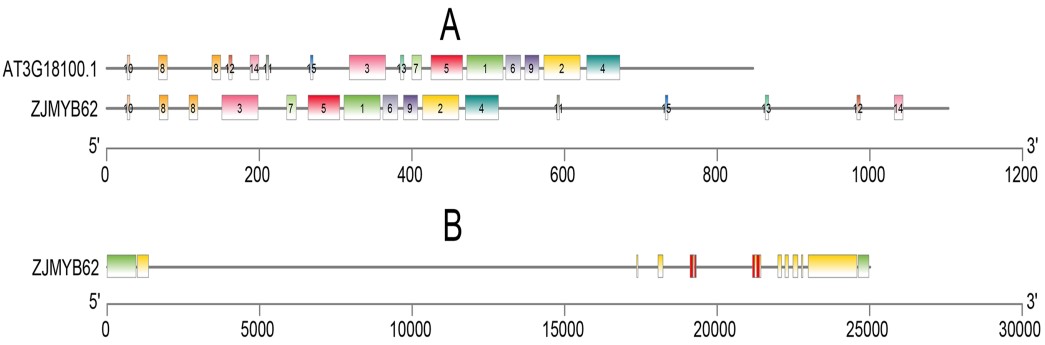

**Figure 6 Motif and gene structure analysis of the 4R-MYB protein.** (A) Distributions of conserved motifs in 4R-MYB genes of *Arabidopsis* and jujube. The motifs of numbers 1–15, are indicated in different colored boxes. The sequence information of motifs had provided in Table S4. (B) Exon–intron structure of 4R-MYB in jujube. The yellow boxes and black lines indicate exons and introns, respectively. Green boxes at the two ends of sequences indicate upstream and downstream regions. The MYB domains are highlighted by red boxes.

We used the MEME program to define the motifs of 4R-MYB (*ZjMYB62*) and *Arabidopsis* (*AT3G18100.1*). Their motifs were very similar, and they all included highly conserved motifs 1–15. Seven groups had highly conserved motifs (Fig. 6A; Table S5), indicating that their sequences were similar. These groups contained four MYB DNA-binding domains. The jujube *4R-MYB* gene had nine introns (Fig. 6B), and the first intron was the longest.

## Distribution on the chromosome and tandem duplicated analysis of *MYB* genes in jujube

Jujube has twelve chromosomes (Zj01–Zj12) (Fig. 7), and 142 of 171 *ZjMYBs* unevenly distributed on chromosomes 1–12 of the jujube genome (Table S2; Fig. 7). From Fig. 7, we can find the higher density of the chromosome genes, and the number of *ZjMYBs* was also relatively large on each chromosome. Zj01 is the largest chromosome and had the most *ZjMYBs*. Three tandem duplicated pairs of jujube *MYB* genes were found on chromosome 2 (*ZjMYB25* with *ZjMYB26*), chromosome 5 (*ZjMYB64* with *ZjMYB65*) and chromosome 11 (*ZjMYB123* and *ZjMYB124*).

## Analysis of *ZjMYB* gene expression from RNA-Seq data

The raw transcriptome sequences from RNA-Seq have been deposited into the NCBI sequence read archive under accessions SRP162927. As shown in Table S7, a total of 46,464,880–50,619,884 raw reads were generated from the 15 libraries. After filtration, a total of 46,244,076–50,550,982 clean reads were obtained from the fifteen libraries with an average Raw Q30 and Clean Q30 base rate of nearly 94%. The overall quality of the sequence data was suitable for further analysis. As revealed by jujube RNA-Seq (Figs. 8A and 8B), the expression levels of different genes at different developmental stages varied. TBtools software yielded a clearer picture of the trend of expression levels of different genes in different stages; the row was also log2 transformed. The reliability of the RNA-Seq data was further validated through real-time quantitative PCR experiments which were carried out on 10 selected MYB genes at five stages (Fig. 8C).

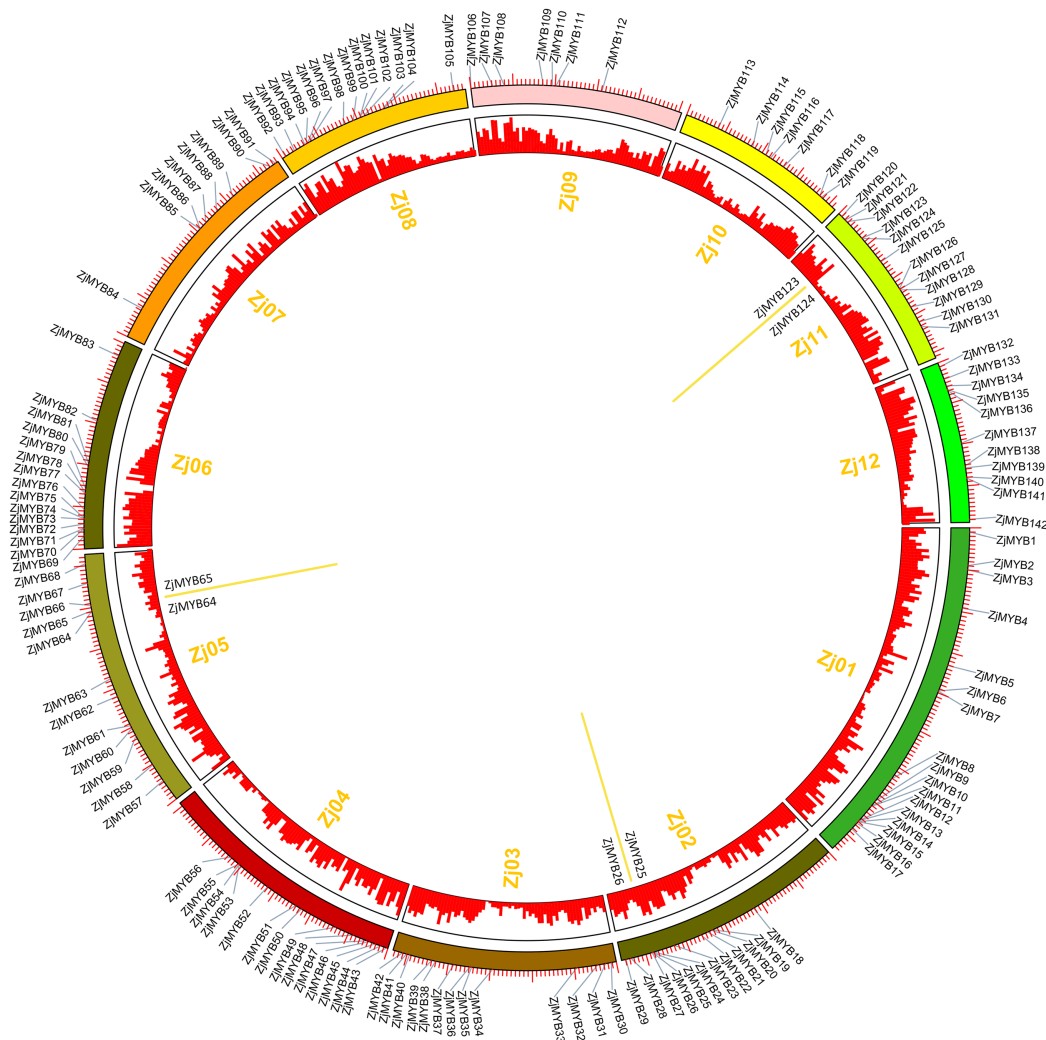

**Figure 7 Chromosomal distribution and gene duplication of jujube MYB superfamily genes.** The inner ring denotes the 12 pseudochromosomes of jujube. The positions of the labels in the optical map are shown by the red bar (a higher bar corresponds to greater density). The outer ring shows the chromosomal distribution of jujube MYB superfamily genes. The scale is five Mb. The three yellow lines inside the circle are the tandem duplicated MYB gene pairs.           

Although extensive analysis of the expression levels of R2R3-MYB genes has been performed (*Kranz et al., 1998*; *Stracke, Werber & Weisshaar, 2001*; *Jia, Clegg & Jiang, 2004*), a similar analysis of 1R-MYB (MYB-related) genes is limited. Interestingly, based on the original FPKM counts in the rectangles, the expression levels of most MYB-related genes (Fig. 8B) were relatively higher than those of R2R3-MYB, R1R2R3-MYB, and 4R-MYB genes (Fig. 8A) in jujube fruit.

Based on the expression profiles of the *ZjMYB* family, the R2R3-MYB, R1R2R3-MYB, and 4R-MYB subfamilies can be divided into nine subgroups, A1–A9 (Fig. 8A), and the MYB-related subfamily can be divided into eight subgroups, B1–B8 (Fig. 8B).

Interestingly, four genes (*ZjMYB126*, *ZjMYB7*, *ZjMYB10*, and *ZjMYB72*) in group B5 and two genes (*ZjMYB92* and *ZjMYB127*) in group A7 similar changes in their gene

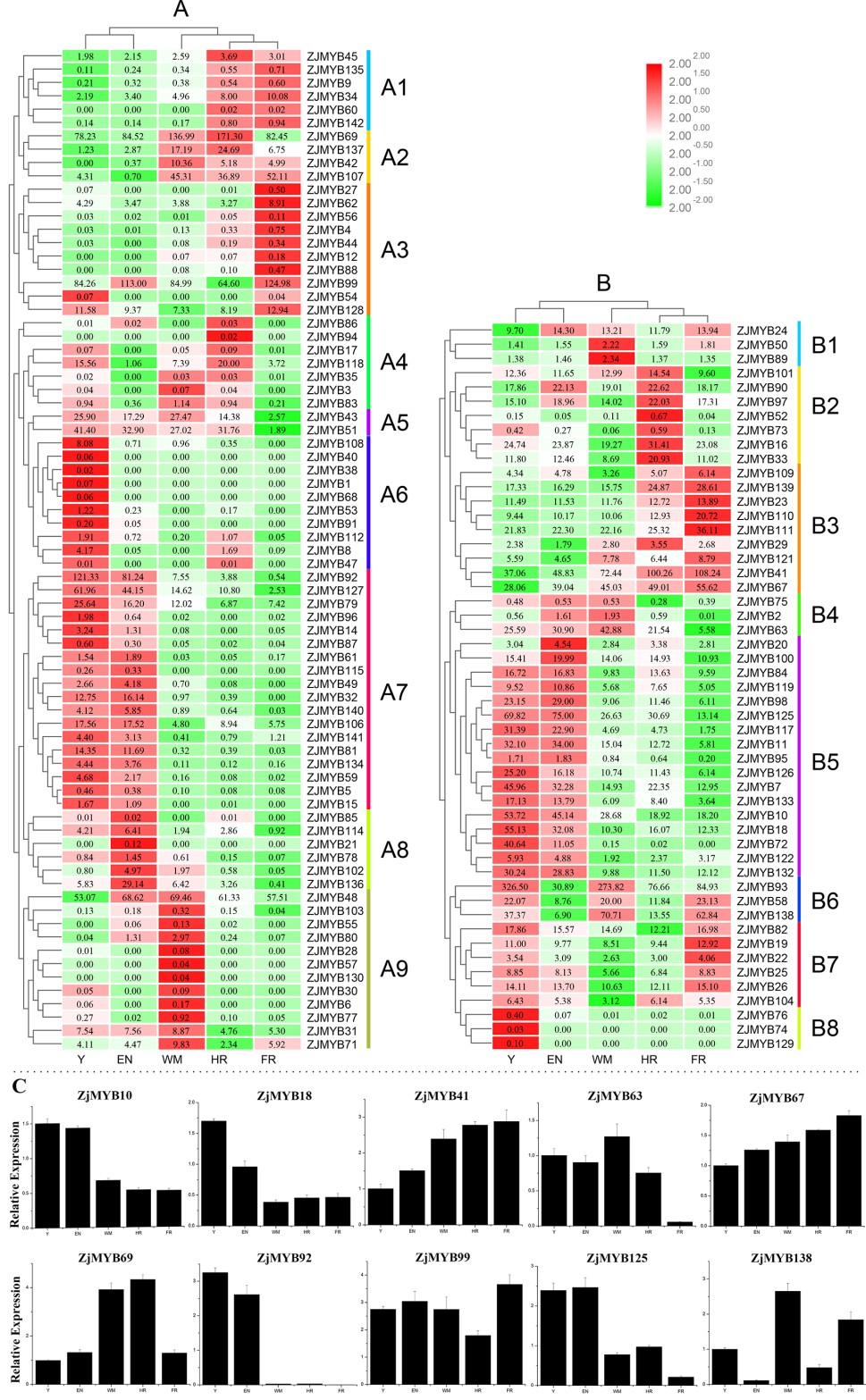

**Figure 8  Heatmap of the expression levels of MYB genes in different developmental stages in jujube fruit and the expression levels of 10 ZjMYB genes by qRT-PCR.** Y, young stage; EN, enlargement stage; WM, white mature stage; HR, half-red stage; and FR, full-red stage. The color scale shown at the top
**Figure 8** (continued)
represents log2-transformed FPKM (fragments per kilobase of exon per million mapped reads) counts. Original FPKM counts are displayed in the corresponding rectangles. Red indicates high expression and green indicates low expression. We eliminated the jujube MYB genes whose expression levels were zero in all stages. (A) Heatmap of the expression levels of 71 R2R3-MYB, three R1R2R3-MYB, and one 4R-MYB genes in different developmental stages in jujube fruit. (B) Heatmap of the expression levels of 44 MYB-related and seven atypical MYB genes in different developmental stages in jujube fruit. A1–A9 and B1–B8 indicate the different subgroups. (C) Expression analysis of 10 MYB genes in jujube by qRT-PCR. The data were normalized to the UBQ gene, and the vertical bars indicate the standard deviation.

expression levels, and all these genes showed a trend from high to low in Y to F. In contrast, the *ZjMYB34* gene in group A1 and three genes (*ZjMYB23*, *ZjMYB41*, and *ZjMYB67*) in group B3 had similar gene expression changes from low to high in Y to FR. According to previous studies, the content of total flavonoids, phenolics, and proanthocyanidins decrease as ripening progresses (*Wu et al., 2012*). Therefore, we predicted that the 10 genes of groups B3, B5, A1, and A7 may possess the function of activating flavonoids and proanthocyanidin biosynthesis.

In A7, most genes displayed high expression at stages Y and EN and had lower expression levels and even no expression in the other stages. Based on this finding, we can infer that the genes in group A7 probably participate in important regulatory functions at the early stages of jujube fruit development. In contrast, group A2 had relatively high expression levels at the late stages (WM, HR, and FR). Compared with the genes in other groups, most genes in groups A1 (except for *ZjMYB34*), A3 (except for *ZjMYB99*), A4 (except for *ZjMYB88*), A6, A8 (except for *ZjMYB136*), A9 (except for *ZjMYB48*), B4 (except for *ZjMYB63*), and B8 had lower expression levels, as well as no expression. The gene expression trend from Y to FR in group B3 and *ZjMYB34* in group A1 was from low to high. The expression level of group A2 increased during the early stages and decreased in the middle stage of jujube fruit development. These results indicated that *MYB* genes may participate in different regulatory mechanisms during jujube fruit development.

## Total flavonoid determination

Consistent with previous studies, the total flavonoid content of jujube decreased with jujube fruit ripening (*Wu et al., 2012*) (Fig. 9). At the FR stage, the content (0.44 ± 0.03 mg RE/g FW) was similar to that in *Z. jujube* cv. Hupingzao (0.47 ± 0.06 mg RE/g FW) (*Kou et al., 2015*). The total flavonoid content was highest in the Y stage and reached 1.23 ± 0.03 mg RE/g FW.

## Correlation analysis

To further explore the associations between the original FPKM counts of transcriptome data of 93 *ZjMYB* genes (containing 42 MYB-related, seven atypical MYB, 40 R2R3-MYB, three R1R2R3-MYB, and one 4R-MYB genes) and total flavonoid contents, we performed a correlation analysis using R software (Fig. 10). A total of 56 MYB genes (23 MYB-related, four atypical MYB, 26 R2R3-MYB, two R1R2R3-MYB, and one 4R-MYB genes) presented significant (<0.05) correlations.

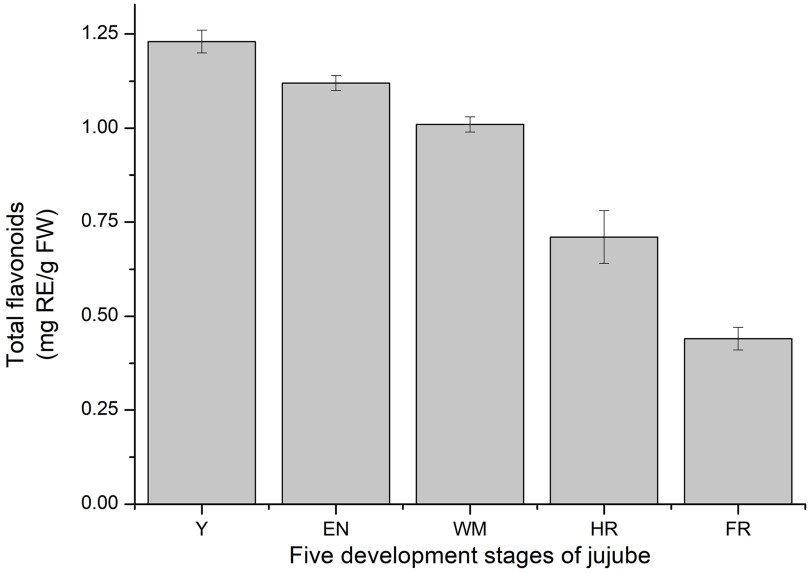

**Figure 9 Total flavonoids content of fruit development stages of jujube.** The results were represented as the mean ± standard deviation. Y, young stage; EN, enlargement stage; WM, white mature stage; HR, half-red stage; and FR, full-red stage.

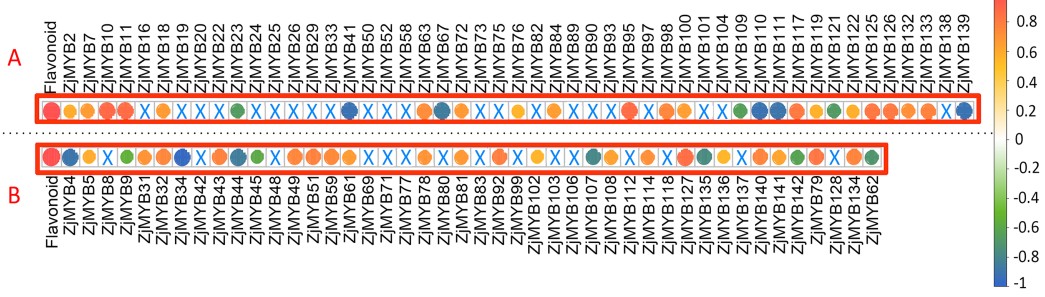

**Figure 10 The correlation analysis of total flavonoid content with ZjMYB gene expression in jujube fruit.** The correlation analysis was performed using the original FPKM counts and the expression levels of the 93 MYB genes and the total flavonoid contents during the five developmental stages in jujube fruit. The "×" symbol indicates that the correlation is not significant at the 0.05 level. Yellow and red circles indicate that the correlation is significant and positive at the 0.05 level, and the green and blue circles indicate that the correlation is significant and negative at the 0.05 level. The correlations between total flavonoid contents and MYB genes are marked in the two red boxes. (A) The correlation analysis of total flavonoid contents with the 42 MYB-related and seven atypical MYB genes. (B) The correlation analysis of total flavonoid contents with the 40 R2R3-MYB, three R1R2R3-MYB and one 4R-MYB genes.

Among these genes, seven MYB-related (*ZjMYB23*, *ZjMYB41*, *ZjMYB67*, *ZjMYB109*, *ZjMYB111*, *ZjMYB121*, and *ZjMYB139*), one atypical MYB (*ZjMYB110*), eight R2R3-MYB (*ZjMYB4*, *ZjMYB9*, *ZjMYB34*, *ZjMYB44*, *ZjMYB45*, *ZjMYB107*, *ZjMYB135*, and *ZjMYB142*), and one 4R-MYB gene (*ZjMYB62*) had a negative correlation with the total flavonoid content. Sixteen MYB-related genes (*ZjMYB2*, *ZjMYB7*, *ZjMYB10*, *ZjMYB11*, *ZjMYB18*, *ZjMYB63*, *ZjMYB72*, *ZjMYB76*, *ZjMYB95*, *ZjMYB98*, *ZjMYB100*, *ZjMYB117*, *ZjMYB119*, *ZjMYB126*, *ZjMYB132*, and *ZjMYB133*), three atypical MYB genes (*ZjMYB84*, *ZjMYB122*, and *ZjMYB125*), 18 R2R3-MYB genes (*ZjMYB5*, *ZjMYB31*, *ZjMYB32*, *ZjMYB43*, *ZjMYB49*,
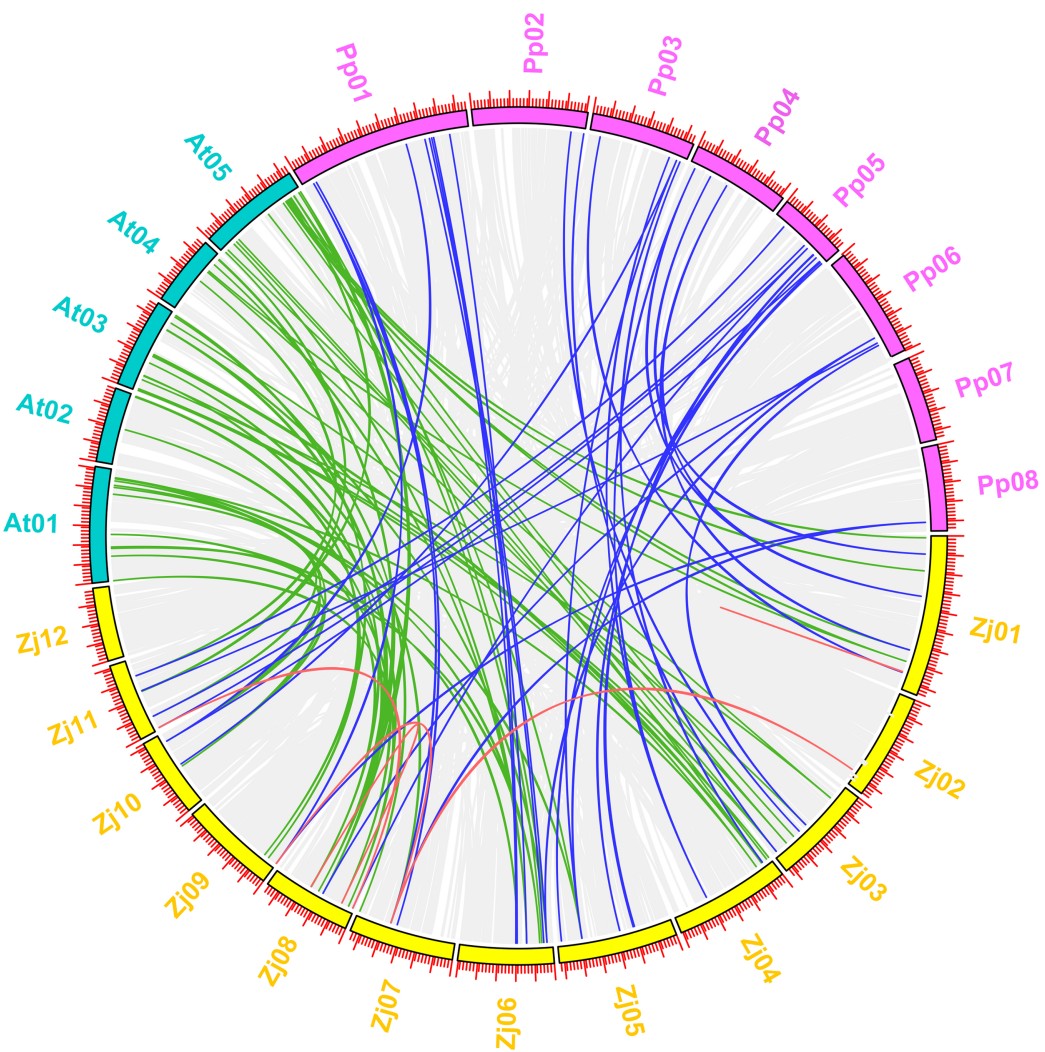

**Figure 11 Synteny analysis of MYB genes between jujube and two other plant species (peach and *Arabidopsis*).** A total of 12 chromosomes of jujube (Zj01–Zj12) and five chromosomes of *Arabidopsis* (At01–At05), as well as eight chromosomes of peach (Pp01–Pp08), were mapped in different colors. The gray lines in the background indicate the collinear blocks within jujube and other plant genomes. The red lines connecting the MYB genes in jujube indicate paralogous MYB gene pairs. The green and blue lines connect the MYB genes in jujube and *Arabidopsis* and in jujube and peach that were orthologous, respectively.

ZjMYB51, ZjMYB59, ZjMYB61, ZjMYB78, ZjMYB81, ZjMYB92, ZjMYB102, ZjMYB108, ZjMYB114, ZjMYB127, ZjMYB136, ZjMYB140, and ZjMYB141), and two R1R2R3-MYB (*ZjMYB79* and *ZjMYB134*) genes were positively correlated with total flavonoid contents.

### ZjMYB gene synteny analysis

*Arabidopsis* is the most important model plant, and many *MYB* genes in *Arabidopsis* have been functionally well characterized. To further determine the phylogenetic mechanisms of the jujube MYB superfamily, we constructed a map based on a syntenic analysis of MYB genes between jujube with other species, including plants from the Rosaceae (peach) and Brassicaceae (*Arabidopsis*) as well as the paralogous *MYB* genes

in jujube (Fig. 11) to predict the function of *ZjMYBs* and analyze the phylogenetic mechanisms of *MYB* genes. This process only highlighted the *MYB* genes for each species. To find more homologous *MYB* gene pairs between jujube and other species, their common gene pairs were all located using syntenic blocks possessing fewer than five homologous gene pairs when we visualized the results using TBtools software.

The five pairs of paralogous *MYB* genes within jujube were thus composed of nine jujube *MYB* genes (Fig. 11 (red line); Table S8). A total of 57 pairs of orthologous *MYB* genes in jujube and *Arabidopsis* were identified among the 171 *ZjMYBs* and 198 *AtMYBs* (Fig. 11 (green line); Table S8). A total of 37 pairs of orthologous *MYB* genes in jujube and peach were identified among the 171 *ZjMYBs* and 256 *PpMYBs* (Fig. 11 (blue line); Table S8). A total of 16 and seven *MYB* genes of jujube showed homology with multiple *MYB* genes from various locations on different chromosomes in *Arabidopsis* and peach, respectively. A total of 14 *ZjMYB* genes of jujube showed the same homology to the *MYB* genes of *Arabidopsis* and peach.

## DISCUSSION

A total of 171 *ZjMYB* genes were identified. Differences in the number of *MYB* genes among diverse species have been studied. For instance, 256 *MYB* genes were found in peach (*Zhang et al., 2018b*), 177 in sweet orange (*Citrus sinensis*) (*Hou et al., 2014*) and 125 in physic nut (*Zhou et al., 2015*). According to previous studies, the MYB family is the largest TF family in jujube (*Liu et al., 2017b*; *Song et al., 2017*; *Shao et al., 2017*; *Zhang et al., 2017*), consistent with previous reports in *Arabidopsis* (*Riechmann et al., 2000*). The *ZjMYB* genes can be divided into five subfamilies, including 58 MYB-related proteins, 99 R2R3-MYB proteins, four R1R2R3-MYB proteins, one 4R-MYB protein, and nine atypical MYB proteins. The R2R3-MYB proteins make up the largest subfamily, consistent with previous reports in peach and *Arabidopsis*. Similar to jujube, a total of 128 R2R3-MYB proteins were identified in peach (*Zhang et al., 2018b*), and 126 were found in *Arabidopsis* (*Yanhui et al., 2006*). Additionally, R2R3-MYB is the largest subfamily of the MYB superfamily. However, this finding is inconsistent with the MYB-related subfamily having the largest number of MYB superfamily genes in sesame (*Mmadi et al., 2017*). These differences are most likely associated with the differences in the evolution of these plants.

In this study, we identified one 4R-MYB protein, which is consistent with previous studies of *Arabidopsis* (*Dubos et al., 2010*), peach (*Zhang et al., 2018b*), and pear (*Li et al., 2016b*). However, the results of this study are inconsistent with those of previous studies reporting four 4R-MYB proteins in Chinese cabbage (*Brassica rapa* ssp *pekinensis*) (*Saha et al., 2016*), two 4R-MYB proteins in upland cotton (*Gossypium hirsutum* L.) (*Salih et al., 2016*), and zero 4R-MYB proteins in sesame (*Mmadi et al., 2017*). In this study, we identified four R1R2R3-MYB proteins, which is consistent with previous studies on peach (*Zhang et al., 2018b*) and tomato (*Li et al., 2016a*), but these results are inconsistent with other previous studies reporting 11 R1R2R3-MYB proteins in Chinese cabbage (*Saha et al., 2016*), five R1R2R3-MYB proteins in sesame
(*Mmadi et al., 2017*), and 15 3R-MYB proteins in upland cotton (*Salih et al., 2016*). These differences are most likely associated with differences in the evolution of these plants.

In previous studies, the first Tryptophan (W) residue of the R3 repeat was replaced by *Leu*cinol hydrochloride (L), Isoleucine (I), and *Phe*nylalanine (F) (*Ogata et al., 1994*; *Du et al., 2012b*). However, in this study, the first Tryptophan (W) of the R3 repeat in jujube was replaced by Tyrosine (Y) and Methionine (M) in addition to the above findings (the back part of motif 2 in Fig. 3); there were two (ZjMYB53 and ZjMYB54) in group C5 (three members) and one (ZjMYB12) in group C12 (one member). The second and third Tryptophan residues in this study were conserved in most R3 repeats, with one (ZjMYB170) occurring in group C34 (two members), and the third Tryptophan (W) residue was replaced by *Phe*nylalanine (F) (the last amino acid of motif 1 in Fig. 3). This finding might be attributed to the loss of residues during jujube evolution or a change in response to changes in the environment, further illustrating the diversity of the MYB domain (*Du et al., 2012a*). Consistent with previous research, the R2 repeat of R2R3-MYB genes also had highly conserved groups of EED and EEE residues (*Zhang et al., 2018b*; *Li et al., 2016b*) (Fig. 3). In jujube, the other residues were highly conserved in the R2 repeat; the eighth and 16th residues behind the first Tryptophan (W) were Leucine (L) and Glycine (G), and the eighth (G) and ninth (L) residues were behind the second Tryptophan (W).

Although many comprehensive analyses of the motif for R2R3-MYB proteins have been performed in previous studies, a similar comprehensive analysis of the motif for the MYB-related subfamily is lacking. The 58 MYB-related (1R-MYB) proteins were divided into six groups, including the CPC-like, TBP-like, I-box-binding-like, R-R-type, CCA1-like, and Chinese jujube-specific groups (Fig. 2). Interestingly, the MYB repeats of CCA1-like proteins are closely related to R-R type MYB repeats (R-R(B)), and I-box-like protein MYB repeats are closely related to R-R-type MYB repeats (R-R(A)) (Fig. 5). We speculate that CCA1-like proteins and I-box-like proteins may be induced by R-R-type gene loss events.

The R2 repeat had highly conserved groups of EED and EEE residues in other plant species, such as peach (*Zhang et al., 2018b*), pear (*Li et al., 2016b*), and tomato (*Li et al., 2016a*). Interestingly, most R1 repeats of MYB-related proteins also included highly conserved groups of EED or EEE residues in jujube (Fig. 5).

The exon numbers of 171 ZjMYB genes ranged from 1 to 20 (Figs. 1C and 2C), indicating the loss and gain of ZjMYB exons during gene evolution, which may account for the functional diversity of the ZjMYB subgroup. As suggested by previous studies, the conserved motifs and intron/exon structures in each subgroup probably play important roles in each group's specific functions (*Zhang et al., 2018b*; *Li et al., 2016b*).

From the Fig. 7, we can see that more MYB genes are located at the two ends of the chromosome than on the middle of the chromosome, and similar patterns were also reported in peach (*Zhang et al., 2018b*) and sweet orange (*Hou et al., 2014*). Despite the lack of description in the article, we can see from a previous paper that the MYB genes of *Arabidopsis* had similar distribution patterns on chromosome 1 and chromosome 5 (*Katiyar et al., 2012*).

The expression of most MYB-related genes (Fig. 8B) was relatively higher than that of R2R3-MYB, R1R2R3-MYB, and 4R-MYB genes (Fig. 8A) in jujube fruit. Thus, we speculated that MYB-related genes probably play a greater role in fruit-related functions. Many of the functions of R2R3-MYB genes have been studied, but the function of MYB-related genes remains unknown. Therefore, we suggest further functional studies of MYB-related genes in the future.

Consistent with previous studies, the total flavonoid content of jujube decreased with jujube fruit ripening (*Wu et al., 2012*) (Fig. 9). According to previous studies, the MYB TF of *Arabidopsis* is involved in many secondary metabolic processes, such as the flavonoid metabolic pathway, glucosinolate biosynthesis, and anthocyanin biosynthesis (*Stracke et al., 2007*; *Bhatia et al., 2018*; *Wei et al., 2015*). For example, the *MdMYBPA1* TF of red-fleshed apple responded to low temperatures by redirecting the flavonoid biosynthetic pathway from proanthocyanidin to anthocyanin production (*Liu et al., 2015*). The MYB superfamily in other plants is also involved in flavonoid/phenylpropanoid metabolism (*Puche et al., 2014*; *Ampomah-Dwamena et al., 2018*; *Wang et al., 2018a*). By correlation analysis between the original FPKM counts of transcriptome data from 93 *ZjMYB* genes (Fig. 10), 56 *MYB* genes presented a significant correlation.

Among the orthologous pairs (Fig. 11) (*ZjMYB108* with *AtMYB12* and *AtMYB11*; *ZjMYB46* with *AtMYB21*, *AtMYB24*, and *AtMYB57*; *ZjMYB88* with *AtMYB15*; *ZjMYB69* with *AtMYB44*; *ZjMYB99* with *AtMYB44* and *AtMYB77*; *ZjMYB65* with *AtMYB28*; *ZjMYB107* with *AtMYB2*; *ZjMYB118* with *AtMYB62*) in jujube and *Arabidopsis*, *AtMYB12* (*AT2G47460*) was expressed at low temperature in a light-dependent manner, and overexpression of the *AtMYB12* gene enhanced flavonoid accumulation (*Bhatia et al., 2018*). *AtMYB11* and *AtMYB12* had a high degree of functional similarity and controlled flavanol biosynthesis (*Stracke et al., 2007*). We speculate that the *ZjMYB108* gene may have the same function. In the correlation analysis results (Fig. 10), *ZjMYB108* gene expression has a positive correlation with flavonoid content; therefore, we further speculate that *ZjMYB108* is an important TF involved in the flavonoid metabolic pathway. *AtMYB21* (*AT3G27810*) is involved in the light signaling pathway (*Shin et al., 2002*). GA (gibberellin) promotes jasmonate (JA) biosynthesis to control the expression of *AtMYB21* (*AT3G27810*), *AtMYB24* (*AT5G40350*), and *AtMYB57* (*AT3G01530*) (*Cheng et al., 2009*). Transgenic *Arabidopsis* expressing *AtMYB15* improved tolerance to cold (*Agarwal et al., 2006*) and drought stress (*Ding et al., 2009*). *AtMYB44* (*AT5G67300*) was rapidly induced by methyl JA in *Arabidopsis*. Transgenic *Arabidopsis* overexpressing the *AtMYB44* gene was more sensitive to ABA and exhibited a markedly increased tolerance to salt and drought stress compared to wild-type plants (*Jung et al., 2008*). *MYB77* (*At3G50060*) regulates auxin signaling processes and auxin concentrations (*Shin et al., 2007*). *AtMYB28* (*AT5G61420*) is involved in aliphatic glucosinolate biosynthesis (*Gigolashvili et al., 2007*). Under drought stress in plants, the *AtMYB2* (*AT2G47190*) protein can function as a transcriptional activator in ABA-inducible gene expression (*Abe et al., 2003*). *AtMYB62* (*AT1G68320*) regulates phosphate starvation responses and gibberellic acid biosynthesis (*Devaiah et al., 2009*). The genes of the MYB orthologous pairs of jujube with *Arabidopsis* may have the same functions.

We predict that the *ZjMYBs* are involved in flavonoid/phenylpropanoid metabolism, the light signaling pathway, auxin signal transduction, and responses to the various abiotic stresses (cold, drought, and salt stresses). Fourteen *ZjMYB* genes of jujube had the same homology in the *MYB* genes of *Arabidopsis* and peach, indicating that the 14 *MYB* genes as well as the orthologous pairs with those 14 genes probably existed before the ancestral divergence of the MYB superfamily.

## CONCLUSIONS

In this study, we performed the first genome-wide detailed analysis of jujube *MYB* superfamily genes, including 171 *MYB* genes (containing 58 MYB-related genes, 99 R2R3-MYB genes, four R1R2R3-MYB genes, one 4R-MYB gene, and nine atypical MYB genes). These 99 R2R3-MYB genes in jujube were divided into 35 groups, C1–C35, and 58 MYB-related genes were divided into the following groups: the R-R-type, CCA1-like, I-box-binding-like, TBP-like, CPC-like, and Chinese jujube-specific groups. The members of the *ZjMYB* gene superfamily in jujube were well supported by additional highly conserved motifs and exon/intron structures. Most R1 repeats of MYB-related proteins that also contained R2 repeats included highly conserved groups of EED and EEE residues in jujube. The composition of the motifs of the 4R-MYB protein in jujube was similar to that in *Arabidopsis*. The exon numbers of *ZjMYB* genes ranged from 1 to 20, indicating loss and gain of *ZjMYB* exons during gene evolution. Three tandem duplicated gene pairs were found on twelve chromosomes of jujube. The analysis of *ZjMYB* gene expression indicated that the different genes had varying expression levels across developmental stages. Interestingly, in this study, the expression of most MYB-related gene expression was higher than that of R2R3-MYB, R1R2R3-MYB, and 4R-MYB genes. In previous studies, the R2R3-MYB subfamily has received more attention. However, in the present study, analysis of the expression of 126 *ZjMYB* genes revealed that MYB-related genes played important roles in jujube development and participated in fruit-related biological processes. The total flavonoid content of jujube decreased with increasing ripening of jujube fruits. A total of 93 expressed genes were identified from the RNA-sequencing data of jujube fruit, and 56 *ZjMYB* members presented a significant correlation with the total flavonoid contents by correlation analysis. Five pairs of paralogous *MYB* genes within jujube were thus composed of nine jujube *MYB* genes. Fourteen *ZjMYB* genes of jujube showed close homology to the *MYB* genes of *Arabidopsis* and peach, indicating that these 14 *MYB* genes as well as their orthologs probably existed before the ancestral divergence of the MYB superfamily. Based on a synteny analysis of *MYB* genes between jujube and *Arabidopsis*, we predict that *ZjMYBs* are involved in flavonoid/phenylpropanoid metabolism, the light signaling pathway, auxin signal transduction, and responses to various abiotic stresses (cold, drought, and salt stresses). We also speculate that *ZjMYB108* is an important TF involved in the flavonoid metabolic pathway. This study provided useful information that may serve as the basis for functional analyses and cloning of *ZjMYB* genes. However, further studies are needed to explore the functions of the *ZjMYB* genes to reveal the molecular regulatory of the mechanisms of these genes in jujube fruit development.

### Funding

This study was supported by the Yunnan Province Ph.D. Scholar Newcomer Award of China and the North Minzu University Research Startup Fund of China. The funders had no role in study design, data collection and analysis, decision to publish, or preparation of the manuscript.

### Grant Disclosures

The following grant information was disclosed by the authors:
Yunnan Province Ph.D. Scholar Newcomer Award of China.
North Minzu University Research Startup Fund of China.

### Competing Interests

The authors declare that they have no competing interests.

### Author Contributions

- Ji Qing conceived and designed the experiments, performed the experiments, analyzed the data, contributed reagents/materials/analysis tools, prepared figures and/or tables, authored or reviewed drafts of the paper, approved the final draft.
- Wang Dawei performed the experiments.
- Zhou Jun authored or reviewed drafts of the paper, approved the final draft.
- Xu Yulan prepared figures and/or tables.
- Shen Bingqi analyzed the data.
- Zhou Fan contributed reagents/materials/analysis tools.

### Data Availability

The raw data are available in a Supplemental File.

### Supplemental Information

Supplemental information for this article can be found online at http://dx.doi.org/10.7717/peerj.6353#supplemental-information.

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
