# Peer review of "Genome-wide characterization and expression analyses of the MYB superfamily genes during developmental stages in Chinese jujube"

_PeerJ, doi:10.7717/peerj.6353_

## Round 0.1 · original submission · Major Revisions

Our reviewers consider that your manuscript unfortunately lacks clarity and should be subjected to language editing and to some re-organization meant to bring your data into a clear focus. You should also provide a link for the raw data of the RNA sequencing. You should also carefully address the other comments regarding the detail of the material and methods section, the conclusions of the correlation analysis and so forth.

When you prepare your rebuttal, provide the full text of all of the reviewers' comments to the initial version of this submission, interspersed with your detailed replies to each point (preferably in a different font, for ease of reading).

PeerJ requests that re-submissions be accompanied by a copy of the manuscript file with highlighted changes. Do not highlight those changes manually: use your word-processor built-in "track changes feature" instead, to compare the initial submission to your modified manuscript.

·

Basic reporting

Where we can find the raw data for RNA sequencing?

Experimental design

no comment

Validity of the findings

In line 394, You can’t get this conclusion only according to the correlation analysis. I don’t think there are so many MYB genes related to the biosynthesis of flavor. In fact, as a large TF family, the MYB family regulates a variety of physiological processes in plants, not only the flavor biosynthesis. I suggest authors refer to the functions of MYB genes in other species that have already been identified.

Additional comments

1. Line 55 “in” should be added before “many plants”
2. Line 120 check the reference
3. Line 196-197, Line 371-378, these sentences should be moved to the discussion part or deleted.
4. Line 418-424 these sentences should be deleted, they have already been described in the introduction part.

Reviewer 2 ·

Basic reporting

Transcription factors are essential on the regulation of gene expression that found in all living organisms. MYB are one of the largest TF gene family present all eukarytes. They have variable function in biological processes inculding development and environment response. Still, many MYBs function are not clear. The authors reports that genome-based identificaton of MYB genes in jujube and showed structure characteristics with RNA-seq data. It could be useful to further effort to understand their function in plants.
However, mostly, the paper to Genome-wide analysis of specific gene family showed that a complete overview of those genes is presented including phylogeny, gene structures, protein motifs/domains, chromosome location and their expression pattern to predict a kind of possible roles in biological processes. these can provide other researchers to understand evolutionary relationships (duplication events, specific subgroups by expanded or contracted) and to further analysis for remained functionally characterizations of genes). But the manuscript in genome-wide study of ZiMYB gene family missed several analysis that I have noted above and showed compelling points. In addition, Too many figure with unclear styles could be ambiguous for audiences to find and understand authors' intent or points. Thorough whole manuscripts, many references were used improperly and unclearly.

Experimental design

Some are a little bit ambigous without details

Validity of the findings

'no comment'

Additional comments

I addressed the major comments like below

Abstract
line 40, expression changes of ZiMYB during tissue and fruit development, that could mean possibility related with those biological processes not clear having function. So. Authors need to tone down.

Introduction
The section of introduction could needs more details, I suggest that the authors explain the history of MYB gene family. The MYB genes are not plant-specific but founded in all eukaryotes.

line 58, addition to Lipsick, 1996, more references need in description for MYB domain
line 66, Katiyar et al., 2012 showed MYB expression pattern under drought stress in Arabidopsis and rice not various biotic and abiotic stresses. Authors should check all citations carefully again.

M&M
Authors need more detail descriptions of fruit periods for samples to do RNA-seq.
e.g. white mature stage’s samples were harvested to how long days after pollination or anthesis etc. or indicate proper references if there are.
Line 120, In reference list, There are two papers represented as Zhang et al 2018. Authors distinguish these on author’s instrument about how to depict references.

The authors performed newly RNA-seq anlaysis for fruit development. But they does not describe detail statistics of RNA-seq data including read depth, read length and quality assessment and so on.
I recommend that authors also need open raw data in public database to be useful future reader.


Result
Line 205, GRAVY, need to describe full name at first
I feel that some figures showed redundancy. e.g. to understand phylogenetic relationship of MYB genes, separated figures 1, 2, 3 and 6 could be combined with simplification and clean up or represented supplementary info. Even though Fig. 4, 5 and 7 showed repeat motifs in different subfamily, I’m not sure that separated figure should be need.

thorough whole manuscripts, many references were used improperly and unclearly.

On Fig. 9, in analysis of gene expression, authors’ insist are based on expression dynamics of MYB genes. But to support these, validation experiments such as qRT-PCR for representative MYB genes not all should be need.
Fig 11, the current figure about the correlation analysis of total flavonoid content with ZjMYB genes of jujube fruit are confused to understand. Plz it need reorganization of figure format by clean up and more concise style

I recommend the position of all ZiMYB gene in pseudomolecules chromosomes could suggest distribution and density of MYB genes and added the results of comparison with peach and Arabidopsis. Additionally, authors need to indicate orthologues within table S1 as shown in Zhang et al, 2018

Syntenny analysis in Fig 12. showed somehow complicated and not clear for audiences to understand their meaning.

Reviewer 3 ·

Basic reporting

A comprehensive analysis of the MYB superfamily members in jujube was done in this study. The authors did a lot of work, including downloading large number of sequences, analysis, making figures and tables.

The authors used clear and unambiguous, professional English throughout the manuscript except for few chinglish.
The authors provided enough Literature references, sufficient field background/context .
The authors shared professional article structure, figs, tables, Raw data.
It is self-contained well with relevant results to hypotheses.

Experimental design

This manuscript is an Original primary research within Aims and Scope of the journal.
Research question is well defined, relevant & meaningful. It is stated well how research fills an identified knowledge gap.
Rigorous investigation was performed, and it reached a high technical & ethical standard.
Methods of this manuscript described well with sufficient detail & information to replicate.

Validity of the findings

Data of this manuscript is robust, statistically sound, and controlled.
Conclusions of this manuscript are well stated, linked to original research question.

Additional comments

Please incline the letters of gene name for the whole manuscript.
please let an english native speaker to revise the manuscirpt before next submission and do not use chinglish.

---

## Round 0.2 · accepted · Accept

We thank you for carefully addressing the reviewer's comments.

# ·

Basic reporting

no comment

Experimental design

no comment

Validity of the findings

no comment

Additional comments

Thank you for addressing my comment. I have no further modifications/suggestions to make.